# Characterization of a Rectangular-Cut Kirigami Pattern for Soft Material Tuning

**Benigno Muñoz-Barron** [1,2], **X. Yamile Sandoval-Castro** [3], **Eduardo Castillo-Castaneda** [2] **and Med Amine Laribi** [4,*]

1 Department of Mechatronics, Tecnológico Nacional de México, Campus Huichapan, Domicilio Conocido Sin número, El Saucillo, Huichapan 42411, Hidalgo, Mexico; bmunoz@iteshu.edu.mx
2 Centro de Investigación en Ciencia Aplicada y Tecnología Avanzada Unidad Querétaro, Mecatrónica, Instituto Politécnico Nacional, Querétaro 76090, Querétaro, Mexico; ecastilloca@ipn.mx
3 School of Engineering and Sciences, Tecnologico de Monterrey, Monterrey 64849, Nuevo León, Mexico; yamile.sandoval@tec.mx
4 Department of GMSC, Pprime Institute CNRS, École Nationale Supérieure de Mécanique et d'aérotechnique, University of Poitiers, 86000 Poitiers, France
* Correspondence: med.amine.laribi@univ-poitiers.fr

**Abstract:** Kirigami is the art of cutting paper to create three-dimensional figures for primarily aesthetic purposes. However, it can also modify the mechanical behavior of the resulting structure. In the literature, kirigami has been applied to modify the material's structural behavior, such as by changing its elasticity, rigidity, volume, or any other characteristic. This article examines the behavior of a pattern of rectangular kirigami cuts on a thermoplastic polyurethane soft material structure and its influence on the mechanical parameters of the macrostructure. The results demonstrate that rectangular kirigami patterns significantly affect the stiffness of the test specimens, changing from 1635 N/m to 4020 N/m. In elongation, there is a variation from 176.6% to 218% by simply altering the height of the rectangular cut. This enables the adjustment of the soft material structure's stiffness based on the geometry of the propagating kirigami cuts.

**Keywords:** kirigami; soft; stiffness

## 1. Introduction

Kirigami and origami have been utilized in engineering for various purposes in the design and production of sensors and actuators. The reported cutting patterns of kirigami can be divided into those with an even distribution on the base material and those with a non-uniform distribution. Similarly, the cuts can be classified into simple and complex cuts. Simple cuts consist of linear cuts distributed along the base material, sometimes forming a combination of straight lines, such as trapezoidal cuts and triangular cuts. Complex cuts are defined as cuts forming curves of greater complexity than a simple straight line, such as circular cuts, spiral cuts, and other shapes of greater complexity. The effects of simple cuts on different base materials and geometries have been analyzed in [1–8] with the aim of creating actuators and sensors that exploit material property tuning and deformations. Refs. [9–13] investigate various cutting patterns using more complex geometries arranged in a regular pattern on the base material, with cuts ranging from micrometric dimensions to a few centimeters.

The applications of kirigami are diverse. For instance, in the field of soft robotics, a triangular-cut kirigami pattern was used as a snakeskin on a pneumatic actuator in [14], increasing the actuator's drag capacity through a mechanism inspired by snake scales. In [15] the author proposed a kirigami–like soft elastomeric skin used to cover a snake-arm robot conformed by compliant vertebrae and controlled with cables. Other patterns were also proposed in [16] that, when cut into a flat material and placed around a pneumatic

actuator, increase its drag capacity. Flexible robots inspired by kirigami utilizing this type of dragging actuation have been developed for medical applications. Ref. [17] demonstrated the development of a robot in which kirigami patterns facilitate the navigation of robots through cavities of the human body under complex conditions and restrictions using hexagonal kirigami patterns. Meanwhile, ref. [18] showcases the development of a flexible robot for medication dosing within the human body, proposing the combined use of two kirigami patterns, one for navigation and another for medication dosing. Other actuators that use kirigami as a key design element can be found in [19–22].

In the realm of sensing applications, kirigami has been widely applied for the development of sensors. The cuts allow for the deformation of the material and its ability to adapt to complex surfaces, enabling the indirect measurement of stresses or deformations produced through the modification of the material's geometry under external forces. Some of the sensors reported in the literature include a heart rate sensor [23], an ECG signal probe [24], angular deformation sensors for robotic actuators [25], and a biocompatible strain sensor [26].

The pattern of simple linear cuts has been widely studied and applied to various base materials [7–32], even in different cut configurations, with most authors approximating the cut thickness to be equal to zero for simplicity of analysis. The applications of this pattern have been varied, including the development of actuators and sensors [30].

The pattern of rectangular cuts has attracted the attention of researchers in various fields, and its study has been conducted from different perspectives and with different configurations. Ref. [33] presents an experimental study of the rectangular cut configuration manufactured on TPU. In their study, the experiment was conducted on a single column of material upon which a rectangular cut was made followed by bonding material with the next rectangular cut. The experimentation considers five cells of rectangular cuts and obtains the parameters of rupture tension, elongation, and stiffness to characterize the behavior of the cutting pattern. Meanwhile, ref. [34] conducted a study using a pattern inspired by rectangular kirigami cuts distributed on TPU material plates manufactured with thicknesses ranging from 2 to 10 mm. In this study, the walls of the test plate containing the rectangular cuts were fully enclosed, which restricts material deformation, although subsequently, the study was conducted on a cylindrical scheme onto which the rectangular cutting pattern was projected. In this study, material stiffness was estimated against wall thickness, and unit deformation was estimated against the reaction force presented by the plate with cuts. Ref. [35], on the other hand, presented an inflatable actuator, which studies various cutting patterns, including the pattern inspired by rectangular cuts. In their research, they created actuators that, in their uninflated form, are configured with the pattern of rectangular cuts. Subsequently, by joining the edges of two plates with rectangular cuts and the central part forming conduction channels, the actuator was subjected to tensile tests and then inflated to characterize its response, mainly its contraction response, thereby achieving behavior similar to that of muscles. The pattern of rectangular cuts has also been studied for sensor development, primarily in its single-column configuration, as shown in the work developed by [36]. In their study, they used a rectangular cut–bonding material–rectangular cut configuration similar to the configuration used by [33], all distributed on a polyamide base as a substrate with synthesized graphene on a copper sheet via low-pressure CVD. In this study, the aim was to characterize the material's resistance behavior as it deforms since the objective was to use the plate as a stress–strain sensor for use on human skin, and no other mechanical tests were conducted in greater depth.

In this study, we propose an investigation into a pattern inspired by rectangular kirigami cuts to characterize properties such as elasticity and stiffness. This investigation involves varying the width of rectangular cuts applied to a 1 mm thick TPU plate. The configuration of the pattern differs from those previously reported, featuring a greater number of rectangular cutting cells distributed horizontally, with free edges following the rectangular cutting pattern. This approach aims to facilitate the development of mechanisms capable of rapidly adapting to specific needs in applications such as robotics, sensors, and actuators based on thermoplastic polyurethane.

## 2. Materials and Methods

### 2.1. Rectangular Pattern Cuts

The proposed pattern of rectangular kirigami cuts is shown in Figure 1. This pattern is defined by the geometry of the rectangular cuts and their distribution on the supporting material. In this work, it is proposed that the cuts have a uniform distribution throughout the entire supporting material, including the edges where the cuts are interrupted by the edge of the base material. The description of the rectangular cuts includes the width (w) and the height (h), with all rectangles having the same dimensions. The distribution of the cuts on the base material is defined by the distance (s) between cuts on the same line and the distance (h1) between lines of cuts. The length (z) of the rectangular cut that coincides with a section of the upper or lower cut is also described.

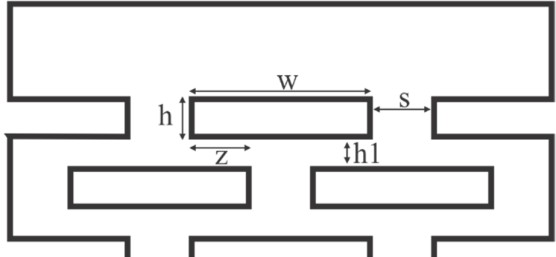

**Figure 1.** Geometrical description of kirigami rectangular cuts.

The behavior of the kirigami cell can be analyzed by discretizing it and examining the behavior of one of the cuts, which can then be used to propagate the deformation phenomenon throughout the entire base material. To model the deformation that occurs along the kirigami base material, a section surrounded by a dotted line, as shown in Figure 2, is proposed to be considered as a representative section of any section along the cell.

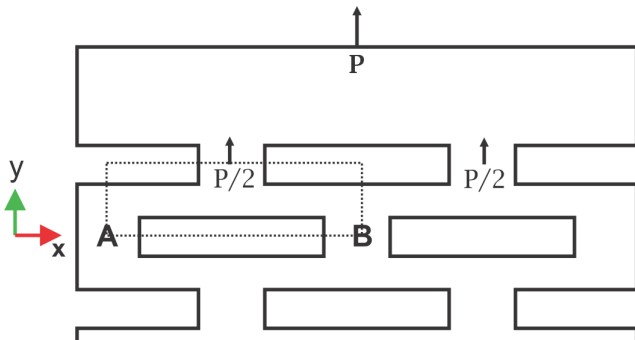

**Figure 2.** Examined segment of the kirigami pattern delineated by a dashed border.

If it is assumed that when the application of force P begins, supports A and B are fixed, as shown on Figure 3; then, it can be considered as if the element forming the section is a small beam with fixed supports. This type of beam allows the ends to be fixed but can undergo rotation, which is ignored in this analysis.

At the begin of the deformation, this structural element is subjected to a force P applied at the center of the beam. By considering the geometry of the beam, in this case, the section described is analyzed from support point A to the center of the beam with a length of L/2. The material is considered to have a thickness $t$ and a width $w$. The equation for the vertical displacement at any point between support A and the center of the beam is valid for the range $0 < x < L/2$. The equation for analyzing deformation is derived through an analysis of shear force and bending moment on the selected segment, as explained on [37]. The development of the analysis leads us to Equation (1), considering the moment of inertia as shown in Equation (2).

$$y = \frac{P}{48EI}\left[4x^3 - 3L^2x\right],\tag{1}$$

$$I = \frac{tw^3}{12},\tag{2}$$

$$y = \frac{P}{Etw^3}\left[x^3 - \frac{3}{4}L^2x\right],\tag{3}$$

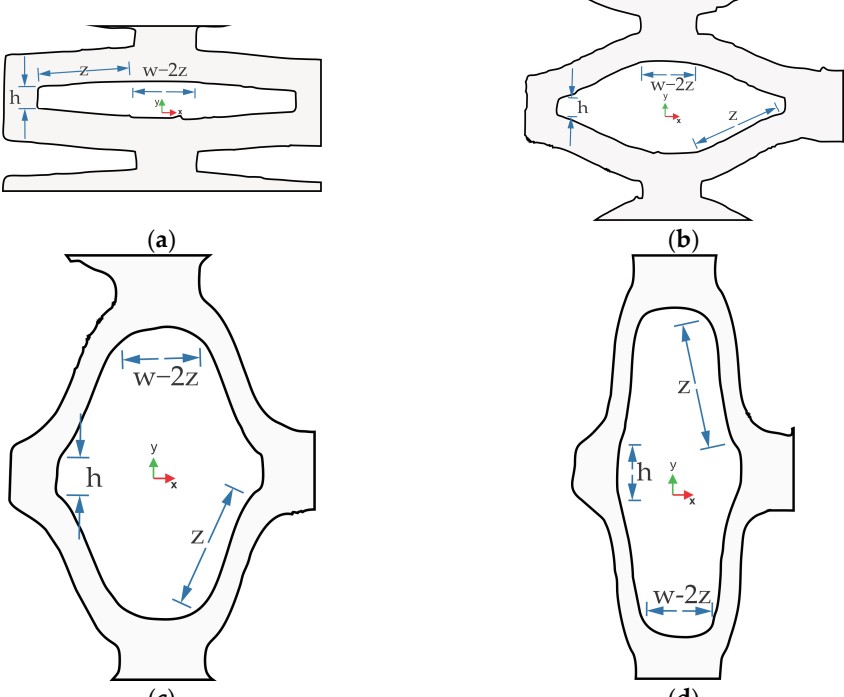

**Figure 3.** Various stages of deformation of the rectangular kirigami cut: (**a**) rectangular cut at the onset of deformation; (**b**) rectangular cut in a second stage of deformation; (**c**) rectangular cut in an advanced stage of deformation; (**d**) rectangular cut in the maximum deformation stage.

Equation (3) indicates the displacement "*y*" of the central point of the beam in Figure 2, which is the point where the displacement is at its maximum. This allows us to identify the maximum deformation of the beam under the established considerations. Once the deformation of the bar-like element is determined, we will illustrate the stages through which the rectangular kirigami cut goes, involving other parameters related to the configuration of the rectangular kirigami cut. Figure 3 shows the different states through which the rectangular cut progresses as deformation advances, as obtained through an analysis of images captured during the experimentation with the specimens. The images are presented as illustrations to more effectively emphasize the points of interest, illustrating the location of various geometric parameters from the initial shape to highlight their impact. In Figure 3a, the rectangular kirigami cut is shown in an initial stage of deformation, where it maintains nearly the same shape, illustrating a single cut since, if the distribution of cuts is uniform along the base material, all cuts not on the boundary exhibit similar behavior.

Figure 3b illustrates the same rectangular cut in a subsequent stage of deformation, still revealing the proportions that constitute the cut, albeit with the rectangular shape having deformed into an octagonal form. In Figure 3c, the kirigami cut is depicted in an advanced stage of deformation, where the rectangular shape has been lost, and a more defined octagonal shape has taken its place, with the sides dependent on the original rectangular geometry, including the cut height "*h*", the cut length "*w*", and the intersecting length "*z*" between the lower and upper cells (defined in Figure 1). Finally, Figure 3d presents the kirigami cut at its maximum stage of deformation, where the rectangular form

has been completely lost, and an octagonal figure has formed, with its sides delineated by the constraints imposed by the original rectangular cut geometry.

From an analysis of Figure 3, we can observe that the displacement "$y$" as presented in Equation (3) adds to the distance "$h$" as we progress toward maximum deformation. Additionally, the maximum deformation is constrained by the distribution of the rectangular cut relative to the upper and lower cuts, as if the distribution of cuts between rows is uniform and symmetrical, and the material between cut lines establishes a limit of "$w-2z$" on the maximum achievable deformation. The maximum vertical deformation attained by a rectangular cut cell transitioning from a height "$h$" to a maximum height "$y_{max}$" is

$$y_{\max} \approx h + 2z \tag{4}$$

as the "$h$" distance increases, it allows for greater freedom of movement for the elements depicted in Figure 2. It is also important to highlight that all rectangular cuts deform uniformly and similarly, except for those at the edges where the uniform distribution over the base material is disrupted. Thus, it becomes possible to sum the contribution of each rectangular cut to estimate the maximum deformation achieved for a plate with rectangular cuts in general.

### 2.2. FEM Simulation

To compare the proposed model, a finite element simulation was conducted using the SolidWorks 2022 software. The TPU material was described using an Ogden hyperelastic model with the following coefficients: $\mu 1 = -30.921$ MPa, $\alpha 1 = 0.508$, $\mu 2 = 10.342$ MPa, $\alpha 2 = 1.375$, $\mu 3 = 26.791$ MPa, and $\alpha 3 = -0.482$. This model considers the effects of 3D printing, as stated in reference [38]. The applied conditions are depicted in Figure 4. The simulation used a triangular-type meshing, and a normal force was applied at one end while an encastre restriction was applied at the other end to simulate a tensile test. In this study, triangular meshing was employed to optimize simulation execution time. The finite element method (FEM) study utilized a Dell G15 5510 laptop equipped with an Intel Core i5-10500H CPU running at 2.50 GHz and 32 GB of RAM memory, operating on Windows 11 Home Single Language version 22H2.

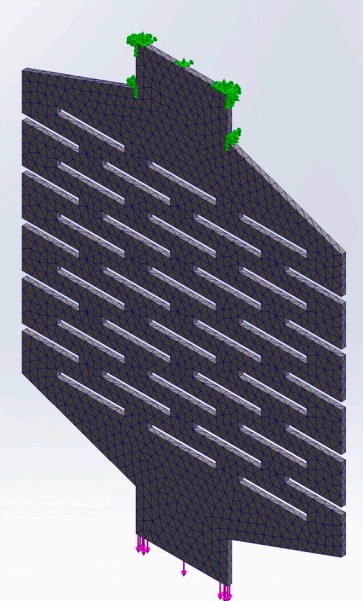

**Figure 4.** Proposed triangular-type meshing over a test specimen described by FEM. The clamping condition is of the encastre type shown in green in the figure, while the applied force, shown in purple, subjects the specimen to tension.

### 2.3. Experimental Setup

To compare the behavior of the kirigami pattern with rectangular cuts, an experimental stress test was conducted on a cell with rectangular cuts. During the tensile test, one end of the test specimen was fixed, and a force was applied to the other end, causing deformation similar to that in the FEM simulation as depicted in Figure 5. The test was carried out using the "LLOYD Instruments Texture Analyzer".

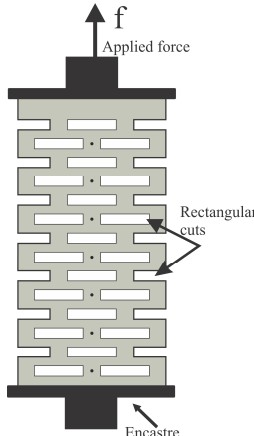

**Figure 5.** Experimental setup using tensile test device.

The test specimen was made of thermoplastic polyurethane (TPU) using a 3D printing process with a Flash Forge Finder printer and their software, Flash Print 5.3.1. The parameters of the printer were set to the default settings for flexible filaments. For the tests conducted using the texture analyzer equipment, test specimens were fabricated while taking into consideration the workspace constraints of the equipment and the clamping requirements of the equipment's jaws. The usable working area of the test specimens measured 50 mm in height by 70 mm in width. The total height of the test specimen was 95 mm, including the material used for supporting the jaws of the tensile test machine. The specimens were manufactured with a thickness of 1 mm, but with variations in the width of the cuts, resulting in specimens with a minimum cut, 1.0 mm cut, and a 1.5 mm cut. Six specimens test were used for each of the proposed variations in the distribution of rectangular cuts. All tests conducted with the equipment were configured with the parameters shown in Table 1.

**Table 1.** Configuration parameters of Flash Forge Finder 3D printer for TPU.

| Velocity | 2.5 mm/s |
| --- | --- |
| Specimen Length | 95 mm |
| Specimen Width | 70 mm |
| Thickness | 1 mm |

The testing equipment allows for the storage of data on applied force, displacement, and time, as well as an estimation of stress.

### 3. Results

The results are presented below; mathematical modeling illustrates the evolution and maximum estimated deformation. The FEM simulation is used to ascertain whether the behavior aligns with experimental outcomes, demonstrating that the considered parameters are accurate and enabling the simulation-based modeling of structures with rectangular cuts. The experimental results are employed to analyze the behavior of cells with rectangular cuts, as well as to validate the findings of the FEM study and the maximum deformation results obtained through mathematical modeling.

### 3.1. FEM Results

FEM analysis demonstrates how deformation occurs when force is applied. Figure 6 shows the transitional state of the kirigami-cut test specimen for one configuration. The analysis indicates that all the cuts experience uniform deformation along the test specimen, except for those at the extreme points. Due to varying constraints, these cuts do not reach maximum deformation, in contrast to the central cuts. Deformations outside the plane of the specimen also occur as the force increases, which are estimated by the FEM simulation, as shown in Figure 6a and observed in experimentation. The FEM simulation also provides us with the results of the patterns' behavior concerning the strain–time simulation shown in Figure 6b and the stresses at the nodes distributed along the central points of the specimen. It is important to note that while the FEM study gives us results per node, the conducted experiment considers the entire specimen as a single mechanical entity.

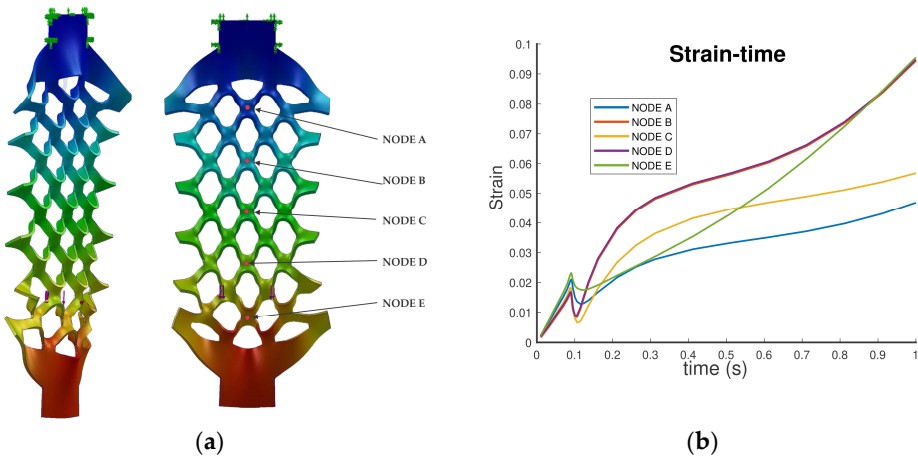

(**a**)                                                                 (**b**)

**Figure 6.** FEM results show the evolution of kirigami cut deformation. (**a**) Test specimen deformation shown at transitional state. (**b**) Strain–time evolution on central nodes of the specimen test simulation.

### 3.2. Experimental Results

The experimental results demonstrate that there is a variation in the behavior of the different specimens in response to changes in the width of the cuts. Similarly, various types of variations can be assessed, such as changes in width, length, and distribution. However, the characterization of the specimens with rectangular cuts allows determining their behavior in response to cut variations. In Figure 7a, the specimen with minimal cuts is shown; these cuts were traced in the 3D printing process and subsequently made directly with a knife. In Figure 7b, the specimen with 1 mm thick cuts is displayed, while in Figure 7c, the specimen with 1.5 mm thick cuts is shown. A total of eighteen test specimens were manufactured for the experimental development, with six specimens for each type of cut and distribution presented. Finally, in Figure 7d, the tension test is shown being conducted on the Lloyd Instruments Texture Analyzer, where it can be observed that one end is secured while a vertical force is applied to the other end, causing displacements and stresses.

Figure 8 displays the final condition before reaching the breaking point of the test specimen. In this state, it is observed that if the distribution of the cuts is uniform, all rectangular cuts deform uniformly along the test specimen, except for those at the extreme points. Therefore, understanding the deformation experienced by a rectangular cut enables the modeling of the behavior of an entire base material with rectangular-cut patterns uniformly distributed on it.

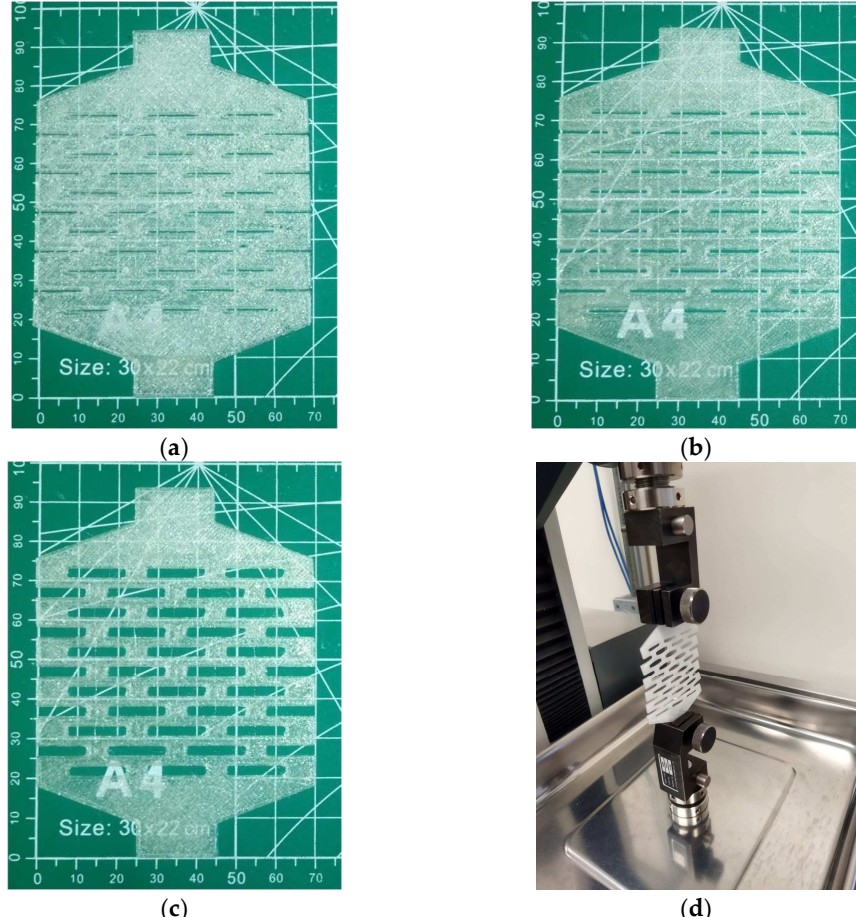

**Figure 7.** Different test specimens with a centimeter-scale background: (**a**) test specimen with almost zero width cuts; (**b**) test specimen with 1 mm width cuts; (**c**) test specimen with 1.5 mm cuts; (**d**) test specimen on texture analyzer.

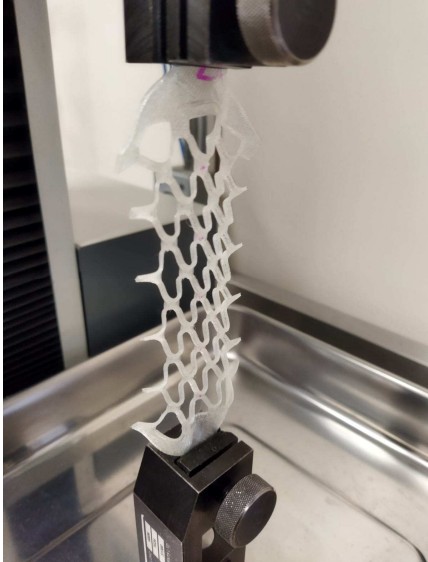

**Figure 8.** Final state for experimental test on test specimen with rectangular 1 mm cuts.

The variation in maximum elongation obtained with each cutting variation is displayed in Figure 9. For the test specimen in which cuts with the greatest width (1.5 mm) were performed, the highest average elongation of 218 percent was achieved. For the specimen

with 1 mm cuts, the average elongation was 188 percent, while for the specimen with the minimum cut, the average obtained elongation was 176 percent. In terms of the variation observed across the tests, it is noted that this variation in final elongation increases with the width of the cut applied to the specimen. Upon reviewing the experimental tests, it was observed that the specimens tended to slip from the grips of the testing device as the displacement increased. Additionally, for tests with a 1 mm wide cut, specimens reached the rupture point at different locations. These variations are believed to be primarily attributed to the manufacturing process.

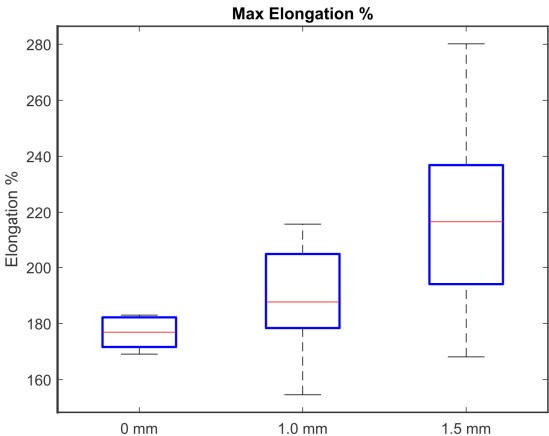

**Figure 9.** Maximum elongation for each test group.

Furthermore, the stiffness associated with the response exhibited the following variation, as indicated by the results estimated by the testing device, and illustrated in Figure 10. For the test specimen with the widest cuts (1.5 mm), the stiffness was 1635 N/m. In the case of the specimen with 1 mm cuts, the average stiffness measured 3264.84 N/m, while for the specimen with the minimum cut, the average stiffness was 4020.29 N/m.

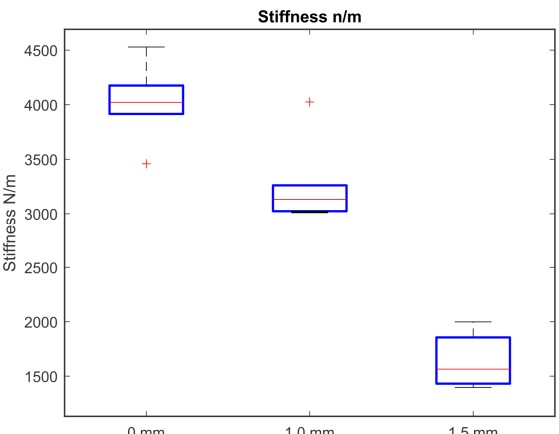

**Figure 10.** Stiffness distribution of each test group. During the data analysis, outliers are displayed for the group with the minimum cut and for the group with a 1 mm cut. This atypical variation (marked with a '+') in the graphs is associated with the manufacturing process of the specimens.

In Figure 11, the stress–strain relationship is shown for a set of experimental tests. It can be observed that each group of specimens exhibits a behavior trend according to the type of rectangular kirigami cut they had. Specimens with minimal cuts show higher stress against the presented deformation, and similarly, the slope of these curves representing stiffness is also higher. Specimens with a 1 mm wide cut reduced the amount of stress for a similar unitary deformation, and the slope of the trend was also lower. Finally, for specimens with a rectangular cut with a width of 1.5 mm, the stress decreased significantly,

and the slope of the trend also decreased compared to that of the previous specimens. It is important to note that in this test, not all specimens reached the breaking point due to the test device's stroke. This can be observed in the vicinity of strain 2, stress 1.5 MPa, where measurements taken by the device are presented once a rupture point occurred in the test specimens with minimal rectangular cuts and with 1 mm wide rectangular cuts. The absence of points related to specimens with 1.5 mm wide cuts is because these specimens did not reach the rupture point during the testing machine's run (trajectories marked with an asterisk on the graph).

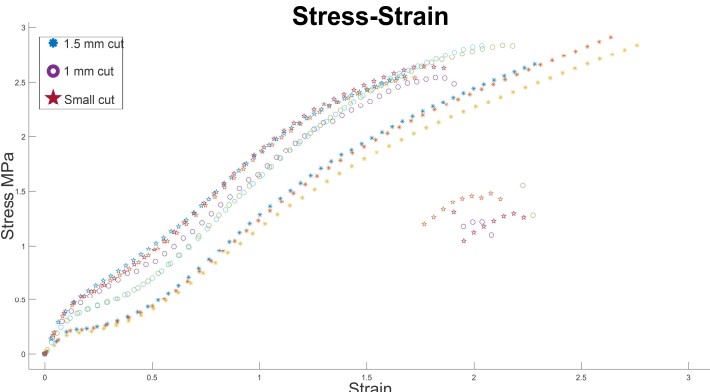

**Figure 11.** Stress–strain relationship for the conducted experimental tests, where each test group exhibits a trend associated with the kirigami rectangular cutting pattern; when the cut is minimal (with a width close to zero), the stiffness is higher, and as the cut increases, forming the rectangular cell, the stiffness tends to decrease.

Regarding the strain behavior shown in Figure 6b, corresponding to the results of the FEM simulation analyzed at the nodes of the test specimen, Figure 12 provides an enlargement of the experimental tension test results. It illustrates that the same phenomenon occurs experimentally at the onset of the test. Initially, the test exhibits some linear behavior, but beyond a certain point (around a strain of 0.1 in Figure 12), there is a sudden drop in the linear behavior. According to our analysis, this drop occurs because out-of-plane deformations start to appear beyond this point. In the conducted experiment, these out-of-plane deformations would symmetrically manifest along the z-plane. For other cuts in the experimental tests, this change is not as pronounced, which we attribute to the resolution limitations of the testing equipment.

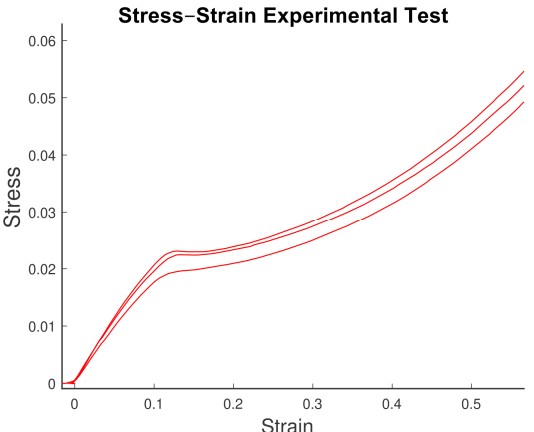

**Figure 12.** Sudden change observed in the stress response of the specimens during the experimental test. Initially, they exhibit linear behavior, until reaching a point where the stress drops rapidly, followed by a further increase but with a different and less linear slope.

## 4. Discussion

The effect of rectangular kirigami cuts was primarily evidenced in the tensile tests conducted on each specimen with rectangular cuts. The variations resulting from changes in the size of the cuts were observed in these tests, and the behavior was validated through finite element method (FEM) simulations. This confirms that the results align with the parameters used in the simulation, ensuring consistency between the experimental and simulated behaviors. The slope of the stress–strain curves corresponds to the stiffness of the test specimen being tested. As seen in Figures 9 and 10, for test specimens with equal cuts, the behavior trend is the same, whereas when comparing test specimens with different cuts, there is a variation in behavior. This allows for the adjustment of variables such as total deformation and test specimen stiffness in a macroscopic behavior.

Figures 9 and 10 provide a summary of the behavior in both maximum extension and stiffness for each test group. It can be observed that as the variation in the width of the rectangular cut increases, the total elongation also increases, but stiffness decreases. For tests where the cut width was greater, the maximum elongation averaged 218 percent, whereas for tests with the minimum cut, the maximum elongation decreased to an average of 176.63 percent, representing an approximate 42% variation due to the increased cut width. Regarding stiffness, tests with a rectangular cut width of 1.5 mm had an approximate average stiffness of 1635 N/m, whereas for tests with a 1.0 mm width cut, the average stiffness increased to 3264.84 N/m. Finally, for tests with the minimum cut, stiffness increased to 4020.29 N/m. In other words, as the cut width increases, stiffness decreases while maximum deformation increases.

Finite element simulation validates the way in which strain occurs along the nodes of the specimen, as well as the behavior of stress response as strain increases. Furthermore, the FEM simulation demonstrates that deformation propagates regularly through the test specimen with rectangular cuts, except for those cuts located at the end of the cell just as it occurs in the experimental stage. In other words, the analysis of a specific test specimen can be used to approximate the behavior of an entire test specimen with "n" cuts distributed along its length, and FEM simulation can predict the behavior of a design using a rectangular-cut pattern using TPU. While FEM simulation validates that the variation in the specimen corresponds to the evolution of deformation observed in the experimental tests, the variation observed in the experimental tests is primarily attributed to the manufacturing process of the specimens. Despite being manufactured under the same conditions, the entropy associated with the 3D printing process is reflected in the behavior of the tensile tests.

These structures are proposed for deployment in both deformable robot development and artificial muscle as potential applications. In Figure 13a, a proposed rehabilitation device is depicted, in which a force is being applied to the test specimen with rectangular cuts. This device can be tailored to the rehabilitation needs of the patient. In Figure 13b, the device is shown when no force is being applied (at rest state).

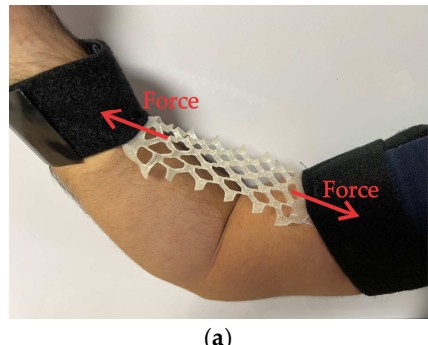 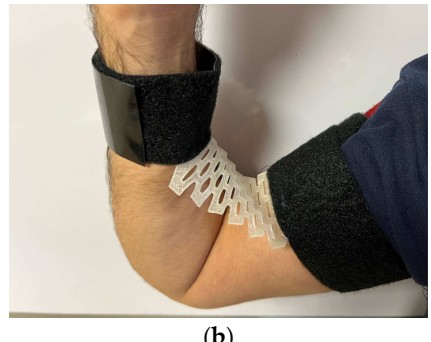

(**a**)　　　　　　　　　　　　　　　　　　　　(**b**)

**Figure 13.** Potential applications of rectangular kirigami-adjusted actuators. (**a**) Device under an axial load produced by the effort of opening the arm; rectangular cells deform under an axial load similar to the case in the study conducted. (**b**) Device in a resting state; cells return to a resting state close to the rectangular shape, albeit with the effect of residual stresses.

Conversely, the rectangular-cut pattern is viable for incorporation into structural elements constructed with TPU, such as the Fin Ray effect-inspired flexible gripper structures depicted in Figure 14. Investigation will assess the potential enhancement of gripper adaptability and grasping by strategically integrating rectangular cuts. Manufacture of these structures employs flexible materials like TPU through 3D printing. The primary advantage of utilizing flexible materials lies in their ability to absorb impact upon contact with other objects, mitigating damage to the colliding entity.

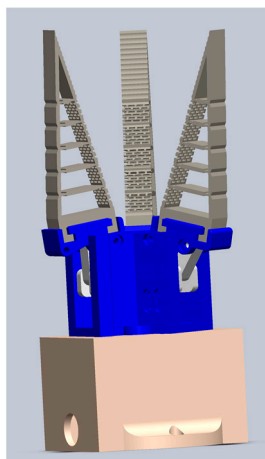

**Figure 14.** The kirigami rectangular-cut pattern will be projected onto TPU structures to adjust their behavior, as seen in Fin Ray-inspired structures for flexible grippers.

A weakness of the study is the lack of analysis regarding the plastic behavior of the test specimens. This is because, during the conducted experimentation, the test specimens were taken to the point of rupture to understand their behavior in response to variations in the width of the cut. Similarly, although this cutting pattern may be employed away from the rupture zone, this study did not provide a fatigue analysis to determine the material's lifespan under the identified conditions.

## 5. Conclusions

The kirigami rectangular-cut pattern affects the stiffness and elongation properties of a TPU specimen as the height of the cut is modified. For the studied configuration, stiffness varies by more than 145% from the minimum cut configuration to the condition where the cut height is 1.5 mm, passing through an intermediate state when the cut is 1 mm. The maximum elongation is also affected, varying by 40% for the TPU material with the described configuration when the cut is at a minimum compared to when the cut is 1.5 mm. This feature, enabled using kirigami rectangular patterns, can be employed to fine-tune the properties of a TPU section to adjust the deformation response to an applied load. The consistent behavior of the test groups confirms that the material performance is uniform, even though the specimens were manufactured through 3D printing rather than strictly through material cutting.

The kirigami rectangular-cut pattern on TPU sheets yields different outcomes compared to those reported by [34,35], who proposed studies similar to the one presented in this research. However, they utilized different configurations in terms of the distribution of rectangular cuts, boundary conditions, and the characteristics of the specimens used. For example, in the study conducted in [34], the effect caused by out-of-plane deformations in the stress–strain curves is not observed due to the lack of interaction with other groups of continuous cells. When the pattern of linear cuts (non-rectangular) has been applied to other materials, a greater elongation has been achieved at the expense of significantly reducing the structure's stiffness. In the presented case using rectangular cuts, a reduction in structure stiffness is also observed, but it is not as drastic. On the other hand, the total elongation is not as substantial as that reported in other studies applying the linear cut

pattern. This may be due to the TPU specimen not necessarily being a sheet but having a thickness of 1 mm. However, for robotic applications, it is undesirable for the structure's stiffness to decrease to levels where it cannot support itself.

Finally, it is worth mentioning that this study explores the behavior of the kirigami rectangular-cut pattern on a TPU specimen using a different configuration regarding the final distribution and boundary conditions of the test specimens. The interest in studying this cutting pattern applied to TPU material stems from its envisioned integration into actuators and sensors operating within more complex systems related to deformable robotics. It is expected that the development of this study will support the modeling and understanding of applications where this pattern is utilized, as discussed in Section 4. To fully characterize the behavior, fatigue tests and analysis of the plastic zone behavior of the constructed actuators and sensors will be conducted. Additionally, the study of new cut configurations will be carried out, considering that small variations in both the distribution and shape of the cuts could result in significantly different behavior.

**Author Contributions:** Conceptualization and validation, B.M.-B., E.C.-C. and X.Y.S.-C.; mathematical modelling B.M.-B. and X.Y.S.-C.; methodology, E.C.-C. and M.A.L.; writing—original draft preparation, B.M.-B., E.C.-C., X.Y.S.-C. and M.A.L.; writing—review and editing, X.Y.S.-C. and B.M.-B.; supervision, X.Y.S.-C., M.A.L. and E.C.-C.; project administration, X.Y.S.-C., M.A.L. and E.C.-C. All authors have read and agreed to the published version of the manuscript.

**Funding:** This research received no external funding.

**Institutional Review Board Statement:** Not applicable.

**Informed Consent Statement:** Not applicable.

**Data Availability Statement:** The data presented in this study are available on request from the corresponding author.

**Conflicts of Interest:** The authors declare no conflicts of interest.

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
