# Peer review of "Characterization of a Rectangular-Cut Kirigami Pattern for Soft Material Tuning"

_applsci, doi:10.3390/app14083223_

Round 1

Reviewer 1 Report

Comments and Suggestions for Authors

Please find comments in the enclosed pdf file.

Comments on the Quality of English Language

Minor editing of English language is required.

Author Response

Dear reviewers,

Dear reviewers, we have addressed the comments you provided regarding the submitted paper “Characterization of rectangular cut kirigami pattern for soft material tuning”. We sincerely appreciate your insightful comments on the review of our article, which are immensely valuable to us as they significantly contribute to the improvement of our work and play a crucial role in the training of new researchers in our field. We value the time and effort dedicated to reviewing our work. In the following sections, we address each of the observations made in detail. Additionally, we have incorporated relevant suggestions to further strengthen and enrich our study. We hope that our responses meet your concerns and expectations, and we are open to any additional suggestions you may have to further enhance the quality of our work. Once again, we sincerely thank you for your contribution to this academic review process. To facilitate the identification of changes made in the manuscript, we have highlighted changes made according to reviewer 1's observations in yellow, while changes based on reviewer 2's observations are marked in cyan.

Reviewer 1

Comments and Suggestions for Authors

Q1. Figure 1, lines 107-109 and Figure 3: how the x and y spatial variables used in (3) are related to h, w, z and h1 cut parameters shown in Figure 1 and 3? Moreover, how the shapes shown in Figure 3 were calculated? Please clearly explain.

Answer: In order to address this observation, the figures were edited to depict the spatial relationship between a frame of reference and the variables describing the geometry of the distribution and the rectangular cut. Furthermore, the following text was added to the body of the article (highlighted in cyan in the article file)

“Figure 6 shows the different states through which the rectangular cut progresses as deformation advances obtained through an analysis of images captured during the experimentation with the specimens, the images are presented as illustrations to emphasize the points of interest more effectively, illustrating the location of various geo-metric parameters from the initial shape to highlight their impact.”

Q2. Figure 2: what does P stands for in Figure 2? I would also suggest to clearly indicate x and y axis directions in that Figure as it is not clear.

Answer: Thank you very much for the observation. To address the same, the definition of variable P has been included, indicating that it is the force transmitted along the test specimen and propagated through the bonding elements between rows of cuts. This has been specified in the text by the following sentences located in the paragraphs as shown below. Additionally, the image has been edited to display the x and y directions describing the deformation of the specimen. The changes in the body of the article are highlighted in cyan.

“If it is assumed that when the application of force P begins, supports A and B are fixed as shown on figure 3, then it can be considered as if the element forming the sec-tion is a small beam with fixed supports.”

“At the begin of the deformation, this structural element is subjected to a force P applied at the center of the beam.”

Q3. Line 162: Figure 4: please clearly describe the mechanical and geometrical constraints used in FEM simulation – the low quality of the image shown in Figure 4 makes it impossible to even guess the constraints. The same applies to the “hex type meshing” which was probably supposed to be visible in this Figure. Moreover, please explain why hex type meshing was chosen. Did it produce better results that simple triangular mesh?

Answer: The figure was enhanced to address this observation. Regarding the specification of the mesh type, this was detailed to establish the simulation conditions under which the study was conducted. This work does not delve into the advantages of the FEM study configuration; however, to have a selection criterion, tests were conducted between the hexagonal and triangular configurations. The main differences were simulation time, and ultimately, the triangular configuration was chosen as it resulted in shorter simulation times with the equipment used. Other simulation conditions, such as the characteristics of the equipment used for the FEM study, were added to the article description. The following modifications were made in the body of the article to emphasize this observation: “In this study, triangular meshing was employed to optimize simulation execution time. The Finite Element Method (FEM) study utilized a Dell G15 5510 laptop equipped with an Intel Core i5-10500H CPU running at 2.50GHz and 32 GB of RAM memory, operating on Windows 11 Home Single Language version 22H2.”

Q4. Figure 6b: please analyze the strain-time data shown in Figure 6b in details – the manuscript is actually lacking such analysis at all. If you do not analyze the strain-time data so what is the purpose to show such dependance in the graph? It is for instance very interesting to know why the strain suddenly drops down at 0,1 s? It seems to be strange that the specimen being stretched actually contracts at some point of its movement. Please also provide a legend in Figure 6b which explains what the colors of the curves are related to – for instance, what is the difference between the green and the blue line? Moreover, the graph shown in Figure 6b suggests that at 0,1s there is a sudden shift from almost linear strain increase to strongly nonlinear behavior. Why such change happens? Please explain.

Answer:  With the aim of explaining this phenomenon, image 6 was edited and a new image 12 was added, contrasting the FEM test result with a group of experimental results; it can be observed that the phenomenon is present in both. According to the analysis conducted, this phenomenon occurs when out-of-plane deformations begin to occur in the specimen with rectangular cuts. To elucidate this phenomenon, the following paragraph was added within the discussion section, although it is also noted that this study does not extensively delve into the out-of-plane deformation phenomenon. The following texts were added to describe (highlighted in cyan in the body of the article):

 “The FEM simulation also provides us with the results of the patterns' behavior con-cerning the strain-time simulation shown in Figure 6(b) and the stresses at the nodes distributed along the central points of the specimen. It is important to note that while the FEM study gives us results per node, the conducted experiment considers the entire specimen as a single mechanical entity.”

“Regarding the strain behavior shown in Figure 6(b), corresponding to the results of the FEM simulation analyzed at the nodes of the test specimen, Figure 12 provides an enlargement of the experimental tension test results. It illustrates that the same phenomenon occurs experimentally at the onset of the test. Initially, the test exhibits some linear behavior, but beyond a certain point (around a strain of 0.1 in Figure 12), there is a sudden drop in the linear behavior. According to our analysis, this drop occurs because deformations out of plane start to appear beyond this point. In the con-ducted experiment, these out-of-plane deformations would manifest along the z-plane symmetrically. For other cuts in the experimental tests, this change is not as pronounced, which we attribute to the resolution limitations of the testing equipment.”

Q5. Figure 9: the deviation of the experimental elongation is quite high for 1,5mm cut width (approx. 30%), moreover, the deviation seems to be strongly related to the cut width – please comment. Please also provide information on the number of test specimens used to prepare graphs shown in Figure 9 (and Figure 10).

Answer: For the experimental development, a total of 18 specimens were used, with 6 for each type of cut presented. To specify this parameter within the text, the following text has been added in the 'experimental setup' section:

“A total of 18 test specimens were manufactured for the experimental development, with 6 specimens for each type of cut and distribution presented.”

Finally, regarding the variation observed in the tests, the videos and the obtained results were analyzed, and the following explanation was included in the article:

“In terms of the variation observed across the tests, it is noted that this variation in final elongation increases with the width of the cut applied to the specimen. Upon review-ing the experimental tests, it was observed that the specimens tended to slip from the grips of the testing device as the displacement increased. Additionally, for tests with a 1 mm-wide cut, specimens reached the rupture point at different locations. These var-iations are believed to be primarily attributed to the manufacturing process.”

Q6. Figure 11: there is a distinguished group of experimental data clustered around strain equal to 2 and stress equal to approx. 1,3 MPa – there are points related to small cuts and 1mm cuts in this area. Please explain the reason for such characteristic area and why points related to 1,5mm cuts are absent in this zone.

Answer: In Figure 11, the behavior of the tensile tests is depicted, with the area referenced where the rupture conditions occurred for the specimens. The absence of points related to the specimens with 1.5 mm cuts is not observed in this zone because these specimens did not reach the rupture point with the displacement of the equipment used. To clarify this phenomenon in the article, the following text has been added:

“It is important to note that in this test, not all specimens reached the breaking point due to the test device's stroke. This can be observed in the vicinity of strain 2 stress 1.5 MPa, where measurements taken by the device are presented once a rupture point occurred in the test specimens with minimal rectangular cuts and with 1 mm-wide rectangular cuts. The absence of points related to specimens with 1.5 mm-wide cuts is be-cause these specimens did not reach the rupture point during the testing machine's run (trajectories marked with an asterisk on the graph).”

Q7. Lines 294-300: in my opinion the manuscript is actually lacking the clear relation between the FEM simulation (in which the material properties were purposedly modelled for 3D printed TPU) and the experimental results. Please provide a clear comparison between numerical FEM and experimental data.

Answer: The discussion section has been enhanced to provide a better comparison between the results. The changes have been highlighted in cyan and have been supplemented with attention to the observations of other reviewers.

Q8. Lines 338-346: please provide direct references to literature in which “different outcome …has been reported so far”.

Answer: Previous works have been referenced for comparison with the results obtained, references 34 to 38 has been added. These authors have applied a similar pattern of rectangular cuts on TPU material; however, the obtained results specifically describe the geometric distribution conditions used, the thickness of the TPU specimen used, and the quantity of cutting cells distributed on the specimen.

To address this observation, the following modification has been made to the body of the article and highlighted in cyan.

“The Kirigami rectangular cut pattern on TPU sheets yields different outcomes compared to those reported by [34] and [35], who proposed studies similar to the one presented in this research. However, they utilized different configurations in terms of the distribution of rectangular cuts, boundary conditions, and the characteristics of the specimens used.”

Reviewer 2 Report

Comments and Suggestions for Authors

The authors have presented a kirigami metamaterial with rectangular cuts and investigated the effect of cut heights using experimental and FEA. The kirigami was created using thermoplastic polyurethane soft material and the results show that the rectangular patterns effect the stiffness of the specimens. Overall, the manuscript has data to support the authors’ conclusions but I believe this work does not add any new contributions to the field and the reasons are described below. Therefore, I do not recommend to publish this work in Applied Sciences.  

Major Comments:

1. My major concern is the novelty of this work. What is the new concept or idea in this work? Is it the rectangular cuts? Is it the polyurethane material for kirigami? Is it modeling the mechanical behavior of the kirigami material through FEA?

2. For rectangular cuts, this geometry has been widely studied and has been shown that by changing the height of the cuts stiffness can be changed.  

a. https://ieeexplore.ieee.org/abstract/document/9479210/authors#authors

b. https://www.nature.com/articles/s41598-018-21479-7

3. For the material, polyurethane has been used to create kirigami structures and studied extensively before.

a. https://www.sciencedirect.com/science/article/pii/S2214860419320123

b. https://ieeexplore.ieee.org/abstract/document/9946378

c. https://www.science.org/doi/full/10.1126/scirobotics.aar7555

4. FEA has been and being utilized to describe the deformation behavior of kirigami structures, what’s new here?

a. https://www.sciencedirect.com/science/article/pii/S1369702119307679

b. https://onlinelibrary.wiley.com/doi/full/10.1002/adma.202001863

5. Further, the authors have not described the in-plane and out-of-plane deformations of the kirigami, and the plastic deformation of the soft polyurethane kirigami structure.

6. The demonstration in Figure 12 does not illustrate any novel concept and Figure 13 would have been great if it were actually created and shown as a gripper.

If the authors can clearly define the novelty of this work and describe how this is different from the previous research efforts in the literature and create the gripper then this work might have some value to the field. In the current state, this work reads like an early work in the field while there are more complex and intricate implementations of the designs with great application demonstrations in the literature.

a. https://www.pnas.org/doi/full/10.1073/pnas.2117649119

b. https://ieeexplore.ieee.org/abstract/document/8978476

7. Finally, in the discussion the authors’ state that this pattern was selected for simplicity in analysis and easy integration into complex systems. The reason all the above examples in literature exist is that the rectangular cuts alone cannot provide the complex deformations needed in soft robotics and 3D actuation systems.

Minor Comments:

Minor grammatical errors throughout the manuscript.

Comments on the Quality of English Language

Minor grammatical errors.

Author Response

Dear reviewers,

Dear reviewers, we have addressed the comments you provided regarding the submitted paper “Characterization of rectangular cut kirigami pattern for soft material tuning”. We sincerely appreciate your insightful comments on the review of our article, which are immensely valuable to us as they significantly contribute to the improvement of our work and play a crucial role in the training of new researchers in our field. We value the time and effort dedicated to reviewing our work. In the following sections, we address each of the observations made in detail. Additionally, we have incorporated relevant suggestions to further strengthen and enrich our study. We hope that our responses meet your concerns and expectations, and we are open to any additional suggestions you may have to further enhance the quality of our work. Once again, we sincerely thank you for your contribution to this academic review process. To facilitate the identification of changes made in the manuscript, we have highlighted changes made according to reviewer 1's observations in yellow, while changes based on reviewer 2's observations are marked in cyan.

Reviewer 2

Comments and Suggestions for Authors

Major Comments:

Q1. My major concern is the novelty of this work. What is the new concept or idea in this work? Is it the rectangular cuts? Is it the polyurethane material for kirigami? Is it modeling the mechanical behavior of the kirigami material through FEA?

Answer: To specify the contribution of this article, the following modifications have been made. Considering the insightful observations made in points 2 and 3, a discussion paragraph has been added analyzing the work done regarding the pattern of rectangular cuts and the materials on which they have been used. Emphasis is placed on the tests performed and their scope. Subsequently, a segment has been added to indicate the difference between the current study and studies found in the literature, which lies in the characterization of the behavior of the rectangular pattern under distribution conditions on the base and manufacturing conditions on the TPU specimen. The constraints under which this study is conducted differ from those published despite using the same cut and analysis tools. The modified text is highlighted in yellow within the body of the article in the introduction section.

Q2. For rectangular cuts, this geometry has been widely studied and has been shown that by changing the height of the cuts stiffness can be changed. 

Answer: In accordance with your insightful comment, the provided references have been reviewed, and an analysis has been included to highlight similarities and differences between the published studies and the current research. While there are similarities in terms of the rectangular cut pattern used, we believe that the experimental conditions, scope, and objectives may differ from the study presented here. To clarify this, a paragraph has been included in the article that analyzes the findings of the referenced studies and how they differ from the study presented here. The integration of the referenced studies and the current study leads to a better understanding of the behavior of rectangular cut patterns in flexible materials. The modified text is highlighted in yellow within the body of the article in the introduction section.

Q3. For the material, polyurethane has been used to create kirigami structures and studied extensively before.

Answer: Regarding the use of polyurethane material, the suggested references have been reviewed. It is considered that the main difference between the studies already published and the one presented here is the configuration in which the Kirigami cuts are distributed on the material. As the results show, there are differences between the published results and those we are presenting, which depend on the thickness of the material used, the boundary conditions of the material plate with cuts, and the distribution of the cuts on the material.

To highlight these differences in the article, the following text has been included in the body of the article in the introduction section.

“In this study, we propose an investigation into a pattern inspired by rectangular Kirigami cuts to characterize properties such as elasticity and stiffness. This investigation involves varying the width of rectangular cuts applied to a 1-millimeter-thick TPU plate. The configuration of the pattern differs from those previously reported, featuring a greater number of rectangular cutting cells distributed horizontally with free edges following the rectangular cutting pattern. This approach aims to facilitate the development of mechanisms capable of rapidly adapting to specific needs in applications such as robotics, sensors, and actuators based on thermoplastic polyurethane.”

Q4. FEA has been and being utilized to describe the deformation behavior of kirigami structures, what’s new here?

Answer: As appropriately noted, the use of FEM for characterizing the behavior of the TPU specimen is a recurrently employed tool in the literature to enhance the analysis of complex structures. In the development of this article, we utilized finite element analysis to model the behavior of the test specimen under the conditions of cut distribution, thickness, and material used. This study reinforces the experimentally conducted tests by demonstrating the same deformation trends experienced by the specimen. We believe that the novelty of the article lies in the characterization obtained from the configuration of cut distribution and boundary conditions of the rectangular-cut specimen under study—a configuration being employed for developing new applications, as illustrated in the prospects.

Q5. Further, the authors have not described the in-plane and out-of-plane deformations of the kirigami, and the plastic deformation of the soft polyurethane kirigami structure.

Answer: In this study, we did not delve into out-of-plane deformations that arise during deformation, as they are not the primary effect sought to be utilized for the subsequent development of applications in deformable robotics and deformable actuators. As for plastic deformations, there is a prospect to utilize equipment that allows testing in this regard to obtain conclusive results. Plastic deformation tests were not included in the presented study since the tests conducted on the specimens reached the rupture zone in most cases. Alongside fatigue testing characterization, this study will be extended to the behavior of plastic deformation in future works. To address this deficiency in the discussion section, the following text has been added regarding this matter (highlighted in yellow in the discussion section within the body of the article):

“A weakness of the study is the lack of analysis regarding the plastic behavior of the test specimens. This is because, during the conducted experimentation, the test specimens were taken to the point of rupture to understand their behavior in response to variations in the width of the cut. Similarly, if this cutting pattern were to be em-ployed away from the rupture zone, this study does not provide a fatigue analysis to determine the material's lifespan under the identified conditions.”

Q6. The demonstration in Figure 12 does not illustrate any novel concept and Figure 13 would have been great if it were actually created and shown as a gripper. If the authors can clearly define the novelty of this work and describe how this is different from the previous research efforts in the literature and create the gripper then this work might have some value to the field. In the current state, this work reads like an early work in the field while there are more complex and intricate implementations of the designs with great application demonstrations in the literature.

Answer: Regarding Figure 13 (on the revised version), a proposed modification to classical structures inspired by the Fin Ray effect is shown, aligning with the prospective applications of the current work. However, this topic is not extensively explored as it is an ongoing study. We believe that for this new study, the variables and parameters to be measured and observed are different and require their own analysis, which should be presented separately. Additionally, efforts are being made in utilizing this pattern in the development of devices for muscle rehabilitation, where other types of variables can also be correlated. Nonetheless, an image depicting the complete concept of the gripper structure being developed is provided to showcase the potential applications of the rectangular cut pattern presented in this article. It is important to highlight that, for the development of this gripper, the distribution analyzed here and the TPU within the thickness range of the specimens fabricated in this study are utilized. Regarding the development of this prototype, we face new challenges in terms of TPU manufacturing capabilities for small objects while trying to maintain the shape and distribution of cuts. 

Q7. Finally, in the discussion the authors’ state that this pattern was selected for simplicity in analysis and easy integration into complex systems. The reason all the above examples in literature exist is that the rectangular cuts alone cannot provide the complex deformations needed in soft robotics and 3D actuation systems.

Answer: Thanks to your insightful comment, the wording of the intended idea has been modified. The objective of the work is to present the characterization of the behavior of the TPU specimen with the rectangular cut pattern for its subsequent use in new sensors and actuators. Considering that the path forward is to first present this characterization and then utilize these results in a more complex structure where we can analyze new variables and include other parameters to construct an intuitive correlation scheme. The wording in the conclusion section has been modified to better convey the idea. 

Round 2

Reviewer 1 Report

Comments and Suggestions for Authors

No additional comments.

Reviewer 2 Report

Comments and Suggestions for Authors

The revisions to the manuscript clearly indicate the novelty of this work and the authors have addressed my concerns. My only suggestion is to improve the resolution of the mechanical data plots but otherwise, I recommend to publish this article.